# CAR-Modified Vγ9Vδ2 T Cells Propagated Using a Novel Bisphosphonate Prodrug for Allogeneic Adoptive Immunotherapy

**DOI:** 10.3390/ijms241310873

**Published:** 2023-06-29

**Authors:** Yizheng Wang, Linan Wang, Naohiro Seo, Satoshi Okumura, Tae Hayashi, Yasushi Akahori, Hiroshi Fujiwara, Yasunori Amaishi, Sachiko Okamoto, Junichi Mineno, Yoshimasa Tanaka, Takuma Kato, Hiroshi Shiku

**Affiliations:** 1Department of Personalized Cancer Immunotherapy, Mie University Graduate School of Medicine, Tsu 514-8507, Mie, Japan; 319ds18@m.mie-u.ac.jp (Y.W.); rieyunahiroshi@clin.medic.mie-u.ac.jp (H.F.);; 2Takara Bio Inc., Kusatsu 525-0058, Shiga, Japan; amaishiy@takara-bio.co.jp (Y.A.);; 3Center for Medical Innovation, Nagasaki University, Nagasaki 852-8588, Sakamoto, Japan; 4Cellular and Molecular Immunology, Mie University Graduate School of Medicine, Tsu 514-8507, Mie, Japan; 5Center for Comprehensive Cancer Immunotherapy, Mie University, Tsu 514-8507, Mie, Japan

**Keywords:** Vγ9Vδ2 T cell, chimeric antigen receptor (CAR), carcinoembryonic antigen (CEA), graft-versus-host disease (GVHD), off-the-shelf, glucocorticoid-induced TNFR-related protein (GITR), tetrakis-pivaloyloxymethyl 2-(thiazole-2-ylamino)ethylidene-1,1-bisphosphonate (PTA)

## Abstract

The benefits of CAR-T therapy could be expanded to the treatment of solid tumors through the use of derived autologous αβ T cell, but clinical trials of CAR-T therapy for patients with solid tumors have so far been disappointing. CAR-T therapy also faces hurdles due to the time and cost intensive preparation of CAR-T cell products derived from patients as such CAR-T cells are often poor in quality and low in quantity. These inadequacies may be mitigated through the use of third-party donor derived CAR-T cell products which have a potent anti-tumor function but a constrained GVHD property. Vγ9Vδ2 TCR have been shown to exhibit potent antitumor activity but not alloreactivity. Therefore, in this study, CAR-T cells were prepared from Vγ9Vδ2 T (CAR-γδ T) cells which were expanded by using a novel prodrug PTA. CAR-γδ T cells suppressed tumor growth in an antigen specific manner but only during a limited time window. Provision of GITR co-stimulation enhanced anti-tumor function of CAR-γδ T cells. Our present results indicate that, while further optimization of CAR-γδ T cells is necessary, the present results demonstrate that Vγ9Vδ2 T cells are potential source of ‘off-the-shelf’ CAR-T cell products for successful allogeneic adoptive immunotherapy.

## 1. Introduction

Chimeric antigen receptor (CAR)-T therapy has achieved considerable success in the treatment of hematologic tumors by using patient derived autologous T cells with αβTCR (αβ T cells). Hence there is a great expectation of using this approach to treat patients with solid tumors. However, CAR-T cell therapy has had disappointing clinical results in solid tumors due to hurdles unique to solid, but not hematologic, tumors including restricted trafficking and limited infiltration into tumors and T cell exhaustion in the tumor microenvironment (TME) [1,2]. Other logistical hurdles of CAR-T cell therapy use in autologous settings include (i) the disease continuing to progress in the patient during the CAR-T therapy manufacturing process (ii) T cell dysfunction of heavily pretreated patients, and (iii) logistical and cost constraints in individualized manufacturing processes [3]. To alleviate these inadequacies, allogeneic CAR-T strategies using a third-party donor derived T cells are desirable. However, graft-versus-host disease (GVHD) that graft’s immune cells recognize and attack the host as foreign, causing tissue damage often seen in after haematopoietic cell transplantation [4] needs to be avoided in these strategies. Although allogeneic CAR strategies have been actively developed using NK cells, it involves time-consuming cell expansion and difficulty in cryopreservation [5].

In human peripheral blood, αβ T cells constitute a majority of T cells which are used as a source of current CAR-T cells. While T cells with Vγ9Vδ2 TCR (γδ T cells) constitute a small (2–5%) fraction of total T cells, they exhibit potent antitumor activity via release of inflammatory cytokines including (1) IFN-γ which inhibits tumor growth and (2) granzymes and perforin, which directly kill tumors [6]. Recognition by Vγ9Vδ2 TCR is independent of major histocompatibility complex (MHC) but dependent on butyrophilin (BTN) 3A1/2A1 [7,8]. As such γδ T cells are not alloreactive and will not cause GVHD in allogeneic transplantation [9]. It indeed has been shown that an adoptive transfer of allogeneic γδ T cells expanded from healthy donors prolonged survival and showed no significant adverse effects such as immune rejection, cytokine storm, or GVHD effects in a patient with solid tumors [10,11]. Another possible advantage of γδ T cells as a source of CAR-T cells is that they recognize tumor cells through phosphoantigens interacting with BTN3A1.2A1 highly expressed in tumor cells but not in normal cells [12]. It has been shown that expanded γδ T cells exert anti-tumor functions, although modest-to-moderate, in early phase clinical trials [13], suggesting genetic modification with CAR expression provides a more beneficial therapeutic effect. Moreover, γδ T cells can be easily expanded using nitrogen-containing bisphosphonates (N-BPs) such as zoledronic acid [14] and cryopreserved without losing immune cell functions [15]. Taken together, γδ T cells appear to have potential as a source of allogeneic ‘off-the-shelf’ CAR-T cells [16]. However, given the innate-like immune cell nature and effector-cell-like metabolic properties of γδ T cells in periphery [17], the persistence/survival and durability of anti-tumor function of CAR-modified Vγ9Vδ2 T cells in vivo remain a major concern.

In the present study, we therefore sought to determine the feasibility of using Vγ9Vδ2 T cells as a source of CAR-T cell products. We employed a novel N-BP prodrug, tetrakis-pivaloyloxymethyl 2-(thiazole-2-ylamino)ethylidene-1,1-bisphosphonate (PTA) which has been shown to stimulate and propagate Vγ9Vδ2 T cells with high purity [18]. The PTA-expanded γδ T cells were modified to scFv specific to carcinoembryonic antigen (CEA) and signaling domains of CD3ζ and CD28, and analyzed for their persistency, localization, phenotypic features, and tumor suppressive activity in a xenograft model using NOG mice.

## 2. Results

*Expansion of γδ T cells from PBMCs utilizing next generation bisphosphonate prodrug PTA in combination with IL-7 and IL-15*; In order to obtain the sufficient number of γδ T cells with high purity, peripheral blood mononuclear cells (PBMCs) from healthy donors were stimulated by a novel prodrug PTA or, as a reference, zoledronate (Zol) and cultured in the presence of Interleukin (IL)-2. Consistent with previously reported results [18], PTA stimulation of PBMCs resulted in a greater number of cells with a higher percentage of CD3^+^Vδ2^+^ T cells when compared to Zol stimulation as previously reported [19] (Figure 1A,B). These PTA-stimulated γδ T cell preparations contained quite a few CD4^+^ or CD8^+^ T cells responsible for GVHDs (Figure 1C). It has been demonstrated that the combination of IL-7 and IL-15 induce a faster and more prolonged proliferation of αβ T cells [20]. Therefore, we next compared the ability of IL-2 and IL-7 plus IL-15 to expand γδ T cells. PBMCs stimulated with PTA and cultured with IL-7 plus IL-15 yielded greater cell numbers as compared to IL-2 (Figure 1D). γδ T cells expanded in the presence of IL-2 or IL-7 plus IL-15 expressed similar levels of costimulatory receptors such as CD28 and GITR (Figure 1E).

*Successful transduction of γδ T cells with a gene encoding CEA-specific CAR and its functional expression*; Having established an optimal condition for stimulation and expansion of γδ T cells, we next examined whether those γδ T cells can be transduced with CAR gene and express functional CAR. To this end, γδ T cells engineered to express a CAR gene composed of anti-CEA scFv F11-39 in the ectodomain and CD28 and CD3ζ signaling endodomains using retrovirus vectors (Figure 2A) [21]. We usually obtained more than 95% of Vδ2+ cells expressing CAR (CEA.CAR-γδ T) (Figure 2B). These CEA.CAR-γδ T cells (1) produced IFN-γ when cocultured with tumors which were expressing various levels of CEA, but did not when cocultured with CEA- tumors (Figure 2C and Appendix A), (2) killed CEA^+^ but not CEA- tumors at various effector (E):Target (T) ratios (Figure 2D) and serially (Figure 2E). Mock-transduced -γδ T cells (Mock-γδ T cells) did not respond to neither CEA^+^ nor CEA- tumors. It is noteworthy that CEA.CAR-γδ T cells produced IFN-γ upon incubation with CEA^+^ (MC32a), but not CEA- tumor (MC38) cell lines which originated from mice and do not express BTN3A1/2A1 [22]. This indicates that CAR signaling by itself, independent of γδ TCR, can induce activation of CEA.CAR-γδ T cells.

*Transferred CEA-specific CAR-γδ T cells suppressed tumor growth in a xenograft mouse model*; To evaluate the therapeutic potential of CEA.CAR-γδ T cells in vivo, we set up experiments where CEA.CAR-γδ T cells were transferred into NOG mice bearing 7-day-old tumor expressing CEA but not GD2 (BxPC-3) (Figure 3A). It has been reported that CAR elicits ligand-independent constitutive signaling (tonic signaling). Varying levels of such signaling induce T cell exhaustion [23] but may also contribute to γδ T cells to control tumor growth, therefore, γδ T cells expressing functional but irrelevant (GD2 specific) CAR (GD2.CAR-γδ T cells) (Appendix A) were also used as a control. As shown in Figure 3B, CEA.CAR-γδ T cells, but not Mock-γδ T cells nor GD2.CAR-γδT cells, suppressed growth of BxPC-3 tumors. The infusion of these γδ T cell preparations did not induce weight loss. In a clinical situation, patients often receive lymphodepletion or myeloablative lymphodepletion prior to CAR-T cell infusion to improve clinical efficacy and/or delay graft rejection in an autologous and allogeneic setting [24,25]. Immunocompetent C57BL/6 mice were treated with fludarabine (Flud), cyclophosphamide (CY) and total body irradiation (TBI) (Appendix A) and we confirmed sufficient lymphodepletion (Appendix A). Then CEA.CAR-γδ T cells were transferred into 8-day-old CEA^+^ tumor (MC32a)-bearing C57BL/6 mice that received the treatment (Appendix A). We found that CEA.CAR-γδ T cells, but not Mock-γδ T cells, suppressed tumor growth (Appendix A).

In tumor-bearing NOG mice, CEA.CAR-γδ T cells gradually reduced in the periphery but gradually accumulated in the tumor (Figure 4). This accumulation of CEA.CAR-γδ T cells in the tumor appeared antigen-dependent since only CEA.CAR-γδ T cells (but not GD2.CAR-γδ T cells nor Mock-γδ T cells) accumulated in the tumors (Figure 4A and Appendix A). We also investigated the expressions of co-inhibitory receptors on γδ T cells in mice bearing CEA+ tumors (Appendix A). We found that γδ T cells irrespective of CAR transduction expressed Tim-3 and LAG-3, which declined gradually over time. It was noted that CEA.CAR-γδ T cells in tumor tissues (but not CEA.CAR-γδ T cells from other tissues nor Mock-γδ T cells from all tissues examined) expressed PD-1 at a later stage. The tumor used in this experiment expressed PD-L1 in vitro and in vivo (Appendix A).

*Gradual loss in tumor reactivity of transferred CEA.CAR-γδ T cells in vivo;* To determine the CEA.CAR-γδ T cells’ durability in maintaining tumor reactivity, single cell suspension of PBMCs, spleen and tumor tissues were cultured in the presence of CEA^+^ (BxPC-3) or CEA- (MIA Paca-2) tumors and Vδ2^+^ cells were analyzed for IFN-γ production by intracellular cytokine staining (ICS). As shown in Figure 5, transferred CEA.CAR-γδ T cells rapidly lost ability to produce IFN-γ upon coculture with CEA+ tumor cells (BxPC-3), irrespective of whether collected from peripheral blood (as PBMCs), spleens, or tumor tissues (as tumor infiltrating lymphocytes (TILs)). CEA.CAR expression on γδ T cells from PBMC remained unchanged by day 20 (Appendix A). It is noteworthy that those CEA.CAR-γδ T cells largely retained the ability to produce IFN-γ upon stimulation with phorbol myristate acetate (PMA) plus ionomycin.

*Co-expression of GITR ligand together with CAR improve anti-tumor function of CEA.CAR-γδ T cells in vivo;* As such CEA.CAR-γδ T cells do not completely lose functionality, we sought to determine whether an additional costimulatory signal may improve CAR-γδ T cell functions in vivo. We employed GITR signaling that provides potent costimulation and may synergize with CD28 signaling for T cell activation [26]. To this end, we transduced a gene encoding a ligand for GITR (GIRTL) to deliver a signal through GITR in addition to CAR to γδ T cells (CEA.CAR.GITRL-γδ T cells). GITR expression was confirmed to be expressed on expanded γδ T cells (Figure 1E). Both genes were successfully transduced in γδ T cells and expressed on the cell surface of γδ T cells (Figure 6A). Co-expression of GITRL significantly enhanced expression of IFN-γ and CD107a, and serial killing function of CEA.CAR-γδ T cells upon coculture with CEA^+^ tumors (Figure 6B,C). Then we compared the ability of CEA.CAR-γδ T cells and those co-expressing GITRL to control tumor growth in NOG mice bearing 11-day-old CEA+ tumor. As shown in Figure 6D, co-expression of GITRL on CEA.CAR-γδ T cells enhanced tumor suppression concomitant with increased infiltration into the tumor and presence in the periphery (Figure 6E,F).

## 3. Discussion

Potent tumor killing function [6], lack of allogenicity [8,9,10], simplicity of expansion [14] and cryopreservation without loss of functionality [15] make γδ T cells an excellent potential source of allogeneic “off-the-shelf” CAR-T cells. Attempts of CAR transduction into γδ T cells expanded using zoledronate have been reported [27,28,29], but the purities of γδ T cell population did not seem sufficient and required further purification before application. In the present study, we demonstrated that γδ T cells stimulated with a novel prodrug PTA [18] achieved greater expansion with high purity as compared to those stimulated with other reagents [19,30]. Typically, PBMCs stimulated with PTA resulted in the cell population containing ~1.5% CD4^+^ and CD8^+^ T cells as compared to ~19% CD8^+^ T cells in zoledronate [19] and 2.7–10.7% CD8^+^ T cells in isopentenyl pyrophosphate [30]. Our results indicate that PTA offers a great opportunity of obtaining a γδ T cell preparation with the least contamination of αβ T cells that cause GVHDs in an allogeneic setting.

As a proof of concept, we employed a second generation CD28 and CD3ζ endodomain-containing CAR and CEA as a target antigen with a favorable expression profile including limited normal tissue expression and broad expression on many solid tumors [31,32]. We confirmed the safety and efficacy of such approach using CEA-transgenic mice transferred with CEA-specific CAR-αβ T cells in our previous study [21]. We could successfully transduce these γδ T cells with CAR in a highly efficient manner without loss of viability. Using the same virus vectors, we have experienced lower transduction efficiency in αβ T cells stimulated anti-CD3 that resulted in ~66% CAR^+^ cells. Previous studies reported by other groups using Zol stimulated γδ T cells also showed lower efficiency in transduction [27,28,29] (<60% CAR^+^). Although these differences in transduction efficiency may be attributable to the difference in multiplicity of infection (MOI; 18.5 or 28.5 in this study), it may be possible that a higher replication rate of γδ T cells induced by PTA contributed to this transduction efficiency and is one of the advantages of using PTA in a γδ T cell preparation.

CEA-specific CAR-γδ T cells respond to various tumor cell lines expressing different levels of CEA even in the absence of signals through γδ TCR and kill tumor cells in a CEA dependent manner. Although it has been reported that serial killing by CAR-γδ T cells has been rarely observed and never been demonstrated directly [33,34], we could demonstrate that CEA.CAR-γδ T cells kill tumor serially, which is an important feature of T cells with efficient tumor control capability [35,36,37,38].

While Themeli et al. [39] demonstrated CAR-γδ T cells almost completely eradicated tumors in an intraperitoneal Rai tumor model, we could not show complete eradication of tumors, which is consistent to other studies [29,40]. In our study, transferred CEA.CAR-γδ T cells seemed to rapidly lose abilities to control tumors in tumor-bearing mice, since ex vivo analysis of these CEA.CAR-γδ T cells revealed that they were unresponsive to tumor at later stage. Roszenbaum et al. proposed that the restricted ability to control tumors by CAR-γδ T cells was due to limited persistence of CAR-γδ T cells and demonstrated that repeated infusion of the CAR-γδ T cells improved anti-tumor effects [29]. However, our results clearly demonstrate that CEA.CAR-γδ T cells accumulated and remained present in tumors, as assessed by fluorescence immunohistochemistry (IHC) staining and flow cytometry with using anti-Vδ2 and/or anti-human CD45 mAbs.

It has been reported that CAR expression on αβ T cells within tumor tissues often eluded the detection by conventional flow cytometry due to the rapid downmodulation and subsequent degradation within the cells upon ligand recognition [41,42], accordingly we could not demonstrate CAR expression on γδ T cells within tumors. Although the proper downmodulation of CAR during ligand binding has been suggested to be required for optimal function of CAR-T cell with αβ TCR [43], recent study has demonstrated that the downregulation of CAR is responsible for the limited CAR-αβ T cell functions in lymphoma and solid tumor models [41,44]. However, sustained CAR expression was observed on Vδ2^+^ cells in PBMC for up to 20 days after the transfer and become unresponsive to tumors, indicating that the loss of CAR expression does not solely account for unresponsiveness of CAR-γδ T cells to tumors.

Although tumors employed in this study express PD-L1 in vivo and in vitro (Appendix A), it is unlikely that co-inhibitory molecules, such as CTLA-4 and PD-1, were involved in the suppression of T cell function in TME [45], since CTLA-4 was not expressed on CEA.CAR-γδ T cells and PD-1 was expressed only at later stage after the transfer. Furthermore, it has been shown that ex vivo expanded γδ T cells are relatively resistant to PD-1 mediated suppression [46]. Expression of Tim-3 and LAG-3, which also play an immunosuppressive role in TME [47], was higher at points in time and rapidly decreased at a later stage when CEA.CAR-γδ T cells became unresponsive to tumors. Taken altogether, these results suggest that the loss of function of CEA.CAR-γδ T cells are not due to limited persistency of infused CEA.CAR-γδ T cells nor loss of CAR expression. Furthermore, immunosuppressive mechanisms in TME may not play a dominant role in the unresponsiveness of CEA.CAR-γδ T cells to tumors.

It has been shown that ligation of GITR delivers a potent costimulatory signal to T cells including γδ T cells [26,48]. Furthermore, not only αβ T cells but also γδ T cells are sensitive to regulatory T cell mediated suppression [48] which is inhibited by GITR signaling induced in regulatory T cells [49,50,51]. In the present study, instead of incorporating a GITR signaling domain in a CAR construct (3rd generation CAR), γδ T cells were co-transduced with CAR consisting CD3ζ and CD28 signaling domain together with a ligand for GITR. This strategy allows GITRL to deliver a GITR signal in CEA.CAR-γδ T cells in order to enhance CAR-T cell function and also deliver a GITR signal to regulatory T cells abundant in tumor microenvironment in order to inhibit their suppressive functions, the latter possibility of which has not been addressed in the present study. Forced expression of GITR ligand on CEA.CAR-γδ T cells enhanced IFN-γ production and serial killing function in vitro and improved in vivo anti-tumor activity associated with increased accumulation in the tumor and enhanced persistency in the periphery. The underlying mechanisms by which GITR signaling enhances CEA.CAR-γδ T cell function remain to be studies further but may involve metabolic changes that support effector functions. [52] It is also possible that GITR signaling promotes CAR-γδ T cells, like αβ T cells [52], to differentiate central memory T cells that maintain effector functions.

In the present study, we mainly employed a subcutaneous xenograft model using immuno-deficient NOG mice. Recent study reported that there were discrepancies in the efficacy of allo-CAR-T cells to control tumors in mice models using NOG mice and fully immunocompetent humanized mice [53]. Furthermore, it also has been shown that orthotopic and ectopic murine models of solid tumors exhibit different tumor microenvironments which determine the efficacy of CAR-T cells [54]. Therefore, future studies are needed by using an orthotopic xenograft model in fully humanized mice to confirm our present results. In conclusion, while further optimization of CAR-γδ T cells is necessary, the present results demonstrate that Vγ9Vδ2 T cells are potential source of ‘off-the-shelf’ CAR-T cell products. The optimization of CAR-T cell products will inevitably include the incorporation of additional measures to ensure the functionality of CAR-γδ T cells in vivo.

## 4. Materials and Methods

*Animals;* NOD/Shi-scid/IL-2Rγnull mice, known as NOG mice, and C57BL/6NcrSlc mice were purchased from the Central Institute for Experimental Animals (Kawasaki, Japan) and Japan SLC Inc. (Shizuoka, Japan), respectively. Mice were fed a standard diet, housed under specific pathogen free conditions, and used at 6–8 weeks of age. All animal experiments were conducted under protocols approved by the Animal Care and Use Committee of Mie University Life Science Center.

*Antibodies and reagents;* The following antibodies and regents were used for cells surface and intracellular staining; FITC-anti-human TCR Vδ2 (Clone: B6), APC-anti-human TCR Vδ2 (Clone: B6), PE-anti-human CD28 (Clone: CD28.2), PE-anti-human GITR (Clone: 621), PE-anti-human CD25 (IL-2R) (Clone: M-A251), PE-anti-human CD127 (IL-7R) (Clone: A019D5), APC-anti-human CD215 (IL-15Rα) (Clone: JM7A4), PE-anti-human CD137 (4-1BB) (Clone: 4B4-1), APC-anti-human CD279 (PD-1) (Clone: EH12.2H7), PE-anti-human CD278 (ICOS) (Clone:C398.4A), PE/Cyanine 7-anti-human CD45RA (Clone: HT100), and as isotype controls, FITC-human IgG1 isotype control (Clone:QA16A12), APC-mouse IgG1, κ (Clone: MOPC-21), PE-mouse IgG1, κ (Clone: MOPC-21), FITC-mouse IgG1, κ, (Clone: MOPC-21), PerCP/Cy5.5-mouse IgG1, κ (Clone: MOPC-21) PE-mouse IgG2a, k, isotype control (Clone: MOPC-173) were from BioLegend; V450-anti-human CD3 (Clone: UCHT1), V500-anti-human CD8 (Clone: RPA-T8), APC-anti-human CD4 (Clone: RPA-T4), V450-anti-human CD45 (Clone: HI30), PE-anti-human CD45 (Clone: HI30) and APC-anti-human CD107a (Clone: H4A3) and Golgi Stop were from BD biosciences; FITC-anti-human CD62L (Clone: DREG-56), PE/Cyanine 7-anti-human CD45RA (Clone: HI100), eFlour 450-anti-IFNγ (Clone: 4S.B3), Alexa488 anti-rabbit IgG, and FITC-anti-Ki67 (Clone: SolA15) were from Invitrogen; FITC-anti-CD66e (CEA) (Clone: REA876) was from Miltenyi Biotec; PE-anti-human GD2 (Clone:14G2a) was from Santa Cruz; PE-conjugated-GITR Ligand (Clone: 109101), Streptavidin APC-conjugated and Streptavidin PE-conjugated were from R&D Systems. Goat-anti-human IgG kappa LC, purchased from MBL. Zoledronate was purchased from NOVARTIS. Tetrakis-pivaloyloxymethyl 2-(thiazole-2-ylamino)ethylidene-1,1-bisophosphohonate (PTA) was synthesized as described [55] and dissolved in DMSO [14]. A final working solution of PTA contained 0.1% DMSO. Modified Yessel’s medium was prepared in house with 35.34 g of IMDM (Gibco) 6.048 g of NaHCO3, 200 mL of human AB serum (Gemcell), 4 mL of 2-aminoethanol (Nacalai Tesque) in PBS, 80 mg of apo-transferrin (Sigma-Aldrich), 10 mL of 1 mg/mL human insulin (Sigma-Aldrich) dissolved in 0.01N HCl, 200 μL of fatty acid mixture containing 4 mg of linoleic acid, 4 mg of oleic acid and 4 mg of palmitic acid in ethanol (all from Sigma-Aldrich), and 20 mL of penicillin/streptomycin solution (Gibco) in 2 L distilled water (MiliQ). After sterilization with 0.22 μm filter, the medium was kept at −20 °C until use.

*Vector construction and preparation of virus solutions;* CAR construct consisting of a sequence identical to a scFv of mAb F11-39 specific to CEA in the VL-VH orientation along with a CD8α hinge, CD28 transmembrane domain, plus CD28 and CD3ζ signaling domains were prepared as described [21] except CD8, CD28, and CD3ζ sequences were replaced with human sequences. A scFv derived from mAb 220-51 [56] was replaced with that from mAb F11-39 in a GD2-specific CAR construct. Construct for GITR ligand contains a sequence of full length human GITR ligand cloned into a pMS3 retroviral vector using Not I and Xho I double digestion sites. The murine stem cell virus LTR was used to drive CAR expression. Virus solutions were obtained as described [57]. Briefly, after transduction into 293T cells (ATCC CRL-3216) with a Retrovirus Packaging Kit Eco (#6160; Takara Bio, Shiga, Japan), the cell culture supernatant was used to transduce PG13 cells ((ATCC CRL-10686)). PG13 cells were transduced with transiently produced ecotropic retroviruses to produce GaLV-pseudotyped retroviruses.

*γδ T cell culture;* Stimulation and expansion of Vγ9Vδ2 TCR T cells were conducted as described with a slight modification [18]. Briefly, human peripheral blood mononuclear cells (PBMCs) from healthy adult donors were obtained by density gradient separation (Ficoll-RaqueTM Plus, Sigma-Aldrich). The cells were then plated at 1.5 × 10^6^ cells/1.5 mL in a well of 24-well plate in YM-AB medium with 1μM PTA. After 24 h, the medium was replaced YM-AB medium with additional supplementation of 300 IU/mL of IL-2 (NOVARTIS) or 25 ng/mL of IL-7 (BioLegend) plus 25 ng/mL of IL-15 (BioLegend) and subsequently expanded to 2 to 5-fold with fresh medium containing IL-2 or IL-7 plus IL-15 at every 2–3 days. All cells were cultured at 37℃ in a 5% CO2 atomospher in a Thermo Forma incubator (Thermo Scientific).

*γδ T cell transduction;* Transduction of γδ T cells with the viral vector was conducted using the RetroNectin-bound virus infection method, wherein virus solutions were preloaded onto Retro-Nectin (#T100A; Takara Bio)-coated wells of a 24-well plate containing 1-mL culture medium as described [58]. Briefly, day 4 and 5 of γδ T cells in stimulation/expansion culture as above were retrovirally transduced with CAR together with or without GITRL at MOI of 18.5 or 28.5 for 10 min under centrifugation at 1000× *g* (HITACH CF16RN, Hitachi, Ltd., Ibaraki, Japan). On day 6, cells were transferred into a T25 Flask with 5 mL YM-AB medium containing IL-7 (25 ng/mL) plus IL-15(25 ng/mL) and subsequently expanded to 2 to 5-fold at every 2–3 days. CAR expression was determined by staining with biotinylated CEA prepared in house using biotin-labelling kit-NH2 (DOJIN, Kumamoto, Japan).

*Tumor cell lines;* Murine colon carcinoma MC38 cells and MC38 cells expressing human CEA (designated MC32a), human pancreatic tumor cell lines BxPC-3, PK9, ASPC1, PCI-66 KP-1N, Panc-1, Capan1, MIA Paca-2, human biliary duct tumor cell lines SSP-25, HuCCT-1, HuH28, TFK1, human gastric tumor cell lines Kato III, MKN45, MKN1, and human periosteal sarcoma Fuji were cultured in RPMI-1640 (WAKO, Osaka, Japan) supplemented with 25 mM HEPES, 10% FCS (Biowest), 2 mM glutamine, 100 U/mL penicillin, and 100 μg/mL streptomycin.

*In vitro assay for CEA.CAR-γδ T cell function;* Cytokine production was analyzed using intracellular cytokine flow cytometry (ICS) or enzyme-linked immunosorbent assay (ELISA) as previously described [57]. Briefly, CEA.CAR-γδ T cells (2 × 10^5^ cells/0.2 mL/well) were co-cultured with tumor cells (2 × 10^5^ cells/0.2 mL/well) in a well of 96-well plate for 6 h in ICS and 24 h in ELISA using a kit for IFN-γ (Invitrogen). The optical absorbance of ELISA wells was measured by sing SpectraMax M2 Plate reader (Molecular Devices). Long-term cytotoxicity was measured using the xCELLigence (ACEA Bioscience) impedance-based assay. Briefly, tumor cells were seeded at 7000 cells/well of a 96-well E-Plate (ACEA Bioscience) and allowed to grow for 18–20 h. CEA.CAR-γδ T cells were then added at various E/T ratios and tumor growth or death as indicated by cell index was monitored up to 72–96 h. Serial killing activity of CEA.CAR-γδ T cells was also assessed by the xCELLigence. Briefly, tumor cells were seeded at 7000, 3500 and 1750 cells/well for 1st, 2nd, and 3rd round co-culture, respectively, and allowed for growth for 18–20 h. Then CEA.CAR-γδ T cells were added at 7000 cells/well and co-cultured for 24 h (initial co-culture). CEA.CAR-γδ T cells from the initial co-cultures were serially transferred to a new well with previously seeded tumor cells at 24-h interval for subsequent rounds of killing (2nd and 3rd) (Appendix A).

*In vivo assay for CEA.CAR-γδ T cell function;* NOG mice were inoculated s.c. with MIA Paca-2 or BxPC-3 (4–5 × 10^6^ cells) on day 0 followed by i.v. injection with CEACAR-γδ T cells or mock transduced γδ T cells (5–10 × 10^6^ cells) on day 7 or day 11. Tumor areas were measured by a microcaliper (A&D-5765A, A&D) using the formula (length × width) at the indicated time points.

*Ex vivo assay for CEA.CAR-γδ T cell function;* Blood, spleen and tumor tissues from NOG mice (n = 4) transferred with CEA.CAR-γδ T cells were collected. PBMCs were separated by Ficoll (Ficoll-Paque PLUS, Sigma-Aldrich, St. Louis, MO, USA). ACK lysing buffer was used for lysis of red blood cells in spleen cells. Tumor tissues were minced in 10 mL HBSS containing 10 mg/mL of collagenase (BioRAD), incubated at 37 °C for 30 min with frequent mixing and filtered through pre-wet 40 µm strainer. Single cell suspensions at 2 × 10^6^ cells/mL for PBMCs, 2–8 × 10^6^ cells/mL for spleens, and 0.3–3 × 10^6^ cells/mL for tumor tissues were cocultured with BxPC-3 or MIA Paca-2 (2 × 10^6^/mL) and subjected for IFN-γ ICS after gating on Vδ2^+^ cells.

*Flow cytometry;* Data on expressions of cell surface molecules and intracellular cytokines were collected by Canto II and LSR Fortessa X-20 (both BD Bioscience). These data were analyzed using FlowJo software (FlowJo, LLC, Oregon).

*Fluorescence IHC staining;* Snap frozen tumor tissues were embedded in OCT compound (SAKURA), and stored at −80 °C until they were sectioned at 3 µm thickness. All sections were stained with fluorescent dye-conjugated anti-CD45, anti-PD-L1 and/or anti-cytokeratin and DAPI. A fluorescence microscope BX53 (Olympus) mounted with DP73 camera (Olympus) was used for imaging of fluorescence IHC staining.

*Statistical analysis;* Data are presented as the mean ± SD where error bars are shown. Statistical analysis was performed by unpaired two-tailed Student’s t-tests using Microsoft Excel. *p* values of less than 0.05 were considered statistically significant. All experiments were conducted more than two times and one of the representative results is shown.

## Figures and Tables

**Figure 1 ijms-24-10873-f001:**
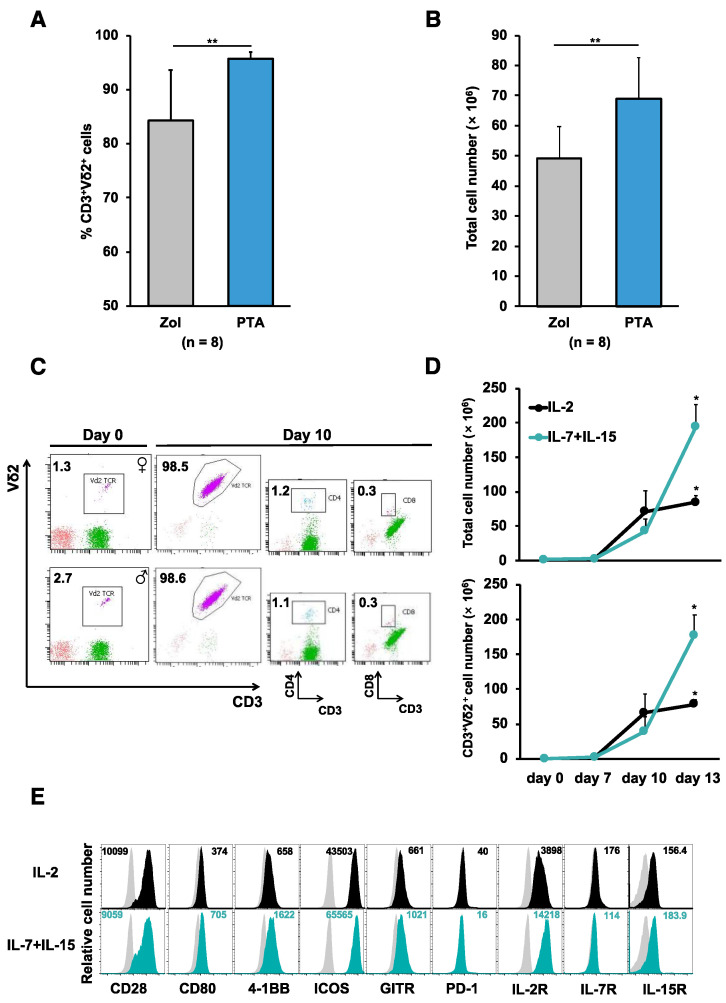
Stimulation and expansion of Vγ9Vδ2 T cells utilizing next generation bisphosphonate prodrug PTA. Frequencies of CD3^+^Vδ2^+^ T cells (**A**) and total cell numbers of PBMCs (**B**) from 8 healthy donors cultured with PTA (1 μM) or Zol (5 μM) in the presence of IL-2 (300 IU/mL) for 11 days. (**C**) Frequencies of CD3^+^ Vδ2^+^, CD3^+^CD4^+^ and CD3^+^CD8^+^ T cells in PBMCs from male and female healthy donors cultured with PTA (1 µM) in the presence of IL-2 (300 IU/mL) for 10 days. (**D**) Numbers of CD3^+^Vδ2^+^ T cells recovered from PBMCs cultured with PTA (1 µM) in the presence of either IL-2 (300 IU/mL) or IL-7 plus IL-15 (25 ng/mL). The results are expressed as a mean ± SD obtained from three independent experiments. (**E**) Cell surface phenotype of CD3^+^Vδ2^+^ T cells from PBMCs cultured with PTA in the presence of IL-2 (black) or IL-7 plus IL-15 (blue) for 10 days. Grey histograms represent staining with isotype control. Numerical values represent Δ mean fluorescence (MFI) (MFI of cells stained with corresponding mAb minus MFI of cells stained with isotype control mAb). Error bars represent SD of the mean. * *p* < 0.05, ** *p* < 0.01.

**Figure 2 ijms-24-10873-f002:**
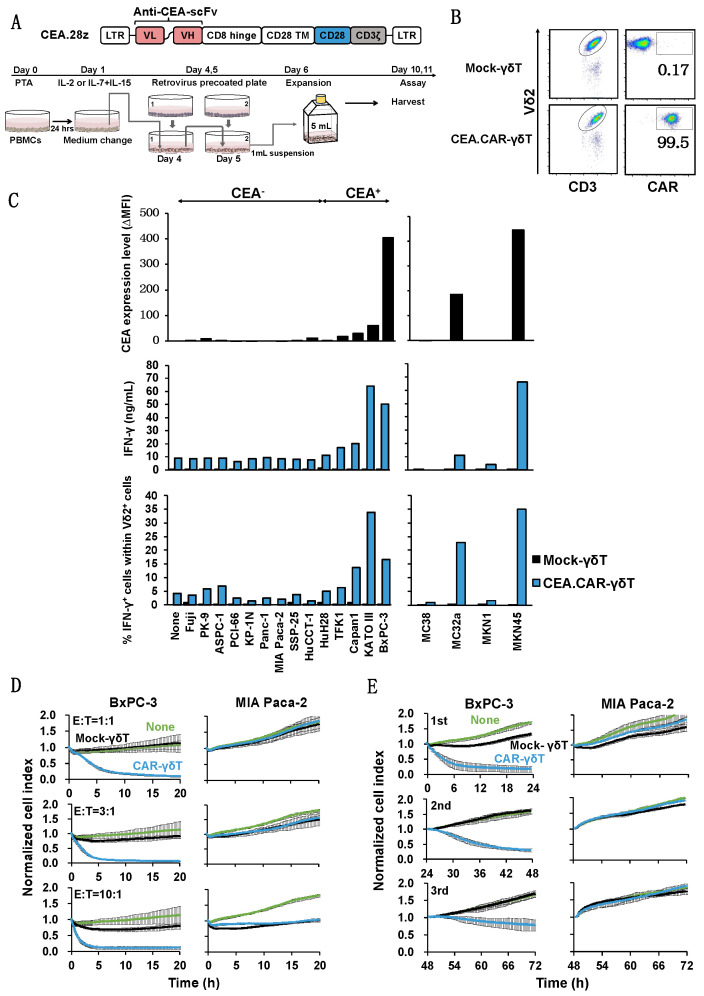
Production of IFN-γ by Vγ9Vδ2 T cells transduced with CEA-specific CAR co-cultured with various tumor cell lines expressing different levels of CEA. (**A**) Schematic representation of a retroviral vector encoding CEA-specific CAR and protocol for preparation of CEA.CAR-γδ T cells. (**B**) Representative of CEA.CAR expression on Vδ2+ T cells on day 10. (**C**) Expression levels of CEA on a variety of tumor cell lines and production of IFN-γ by CEA.CAR-γδ T cells co-cultured with corresponding human and murine tumor cell lines as described in MATERIALS AND METHODS. Expression levels of CEA were expressed as delta changes MFI (ΔMFI = MFI of cells stained with anti-CEA minus MFI of cells stained with isotype control mAb). A representative result of 2 independent experiments is shown. (**D**) Cytotoxic activity of CEA.CAR-γδ T cells against CEA^+^ BxPC-3 and CEA- MIA Paca-2 at E/T ratios of 1:1, 3:1 and 10:1 was analyzed using an xCELLigence impedance-based real-time cell analyzer. Green lines; untreated mice (None), Black lines; the mice treated with Mock-T cells, Blue lines; the mice treated with CAR-γδ T (**E**) Serial killing activity of CEA.CAR-γδ T cells was measured as above with 3:1 E/T ratio up to 3 rounds. Each killing was monitored up to 24 h. A representative result of 3 independent experiments is shown. Error bars represent SD of the mean.

**Figure 3 ijms-24-10873-f003:**
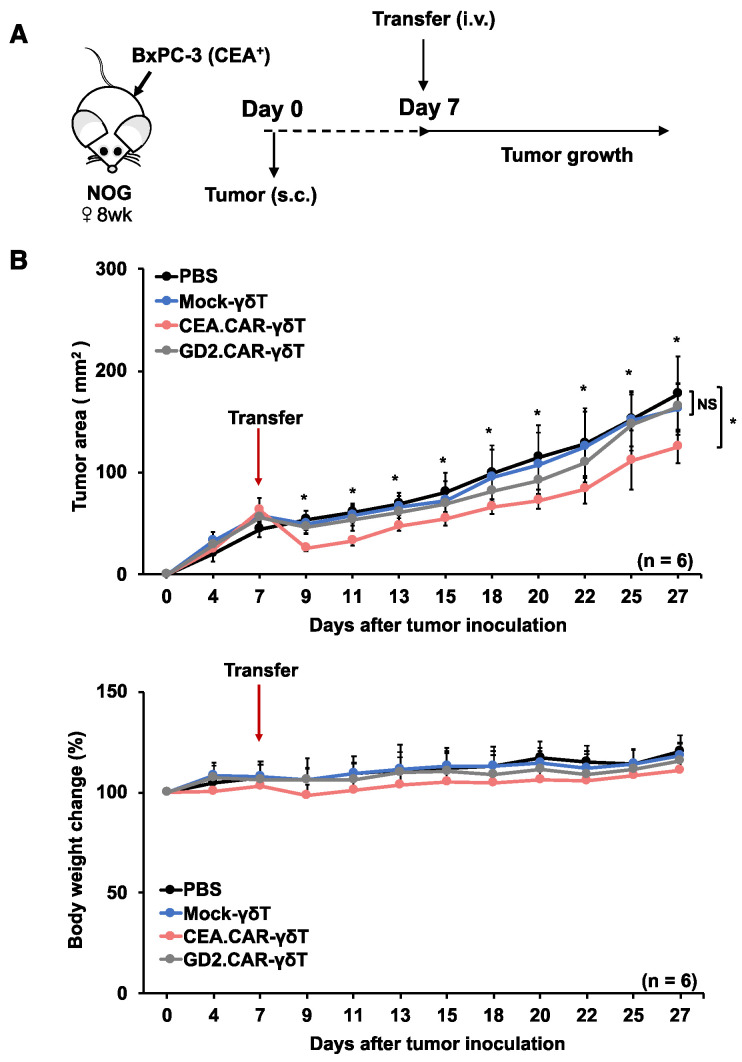
Effective but transient tumor growth control by adoptive transfer of CEA.CAR-γδ T cells. (**A**) Schematic representation of the adoptive transfer experiment using NOG mice. (**B**) Tumor growth curves of BxPC-3 in NOG mice and body weight changes of tumor bearing NOG mice (n = 6) transferred with CEA.CAR-γδ T cells, GD2.CAR-γδ T cells or Mock-γδ T cells. NOG mice were inoculated s.c. with BxPC-3 (5 × 10^6^ cells) (on day 0) followed by i.v. injection with CEA.CAR-γδ T cells, GD2.CAR-γδ T cells or Mock-γδ T cells (5 × 10^6^ cells) (on day 7). Tumor areas were measured by a caliper using the formula (length × width) at the indicated time points. Error bars represent SD of the mean. * *p* < 0.05. A representative result from 3 independent experiments is shown.

**Figure 4 ijms-24-10873-f004:**
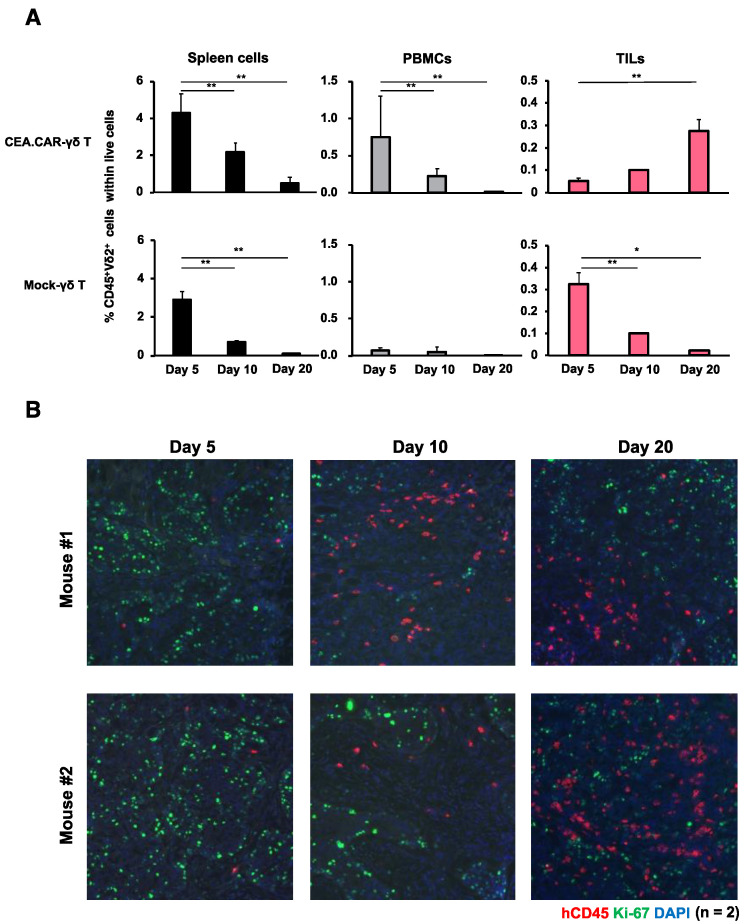
Accumulation and persistence of CEA.CAR-γδ T cells within tumor tissues. (**A**) Spleens, PBMCs (n = 4 at each time point) and tumor tissues (n = 2 at each time point) of tumor-bearing NOG mice transferred with (5 × 10^6^) CEA.CAR-γδ T cells or Mock-γδ T cells were collected at 5 days, 10 days and 20 days after the transfer, and subjected to flow cytometry analysis. The Percentages of human CD45^+^Vδ2^+^ cells were obtained using total live cells based on FSC and SSC profiles. (**B**) Fluorescence IHC analysis of tumor tissues collected in each time point (n = 2) stained with anti-human CD45 (red), Ki-67 (green) and DAPI (blue). A representative result of 3 independent experiments is shown. Error bars represent SD of the mean. * *p* < 0.05, ** *p* < 0.01.

**Figure 5 ijms-24-10873-f005:**
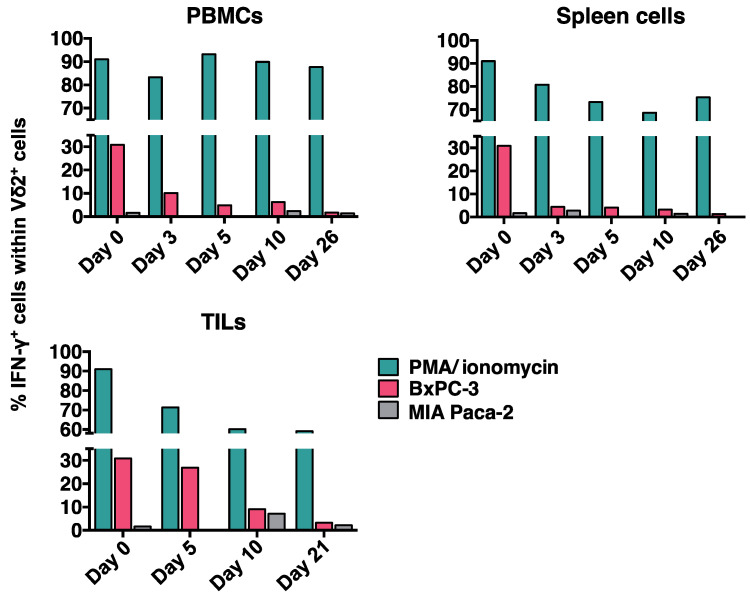
IFN-γ production of CEA.CAR-γδ T cells recovered from tumor-bearing NOG mice. Spleens, PBMCs (n = 4 at each time point) and tumor tissues (n = 2 at each time point) from tumor (BxPC-3)-bearing NOG mice transferred with CEA.CAR-γδ T cells were pooled, co-cultured with fresh CEA+ (BxPC-3) or CEA- (MIA Paca-2) tumors and subjected to IFN-γ ICS after gating on Vδ2+ cells as described in MATERIALS AND METHODS. Cells stimulated with PMA (100 nM) plus ionomycin (1 µg/mL) served as a control. Results are combined from more than three independent experiments.

**Figure 6 ijms-24-10873-f006:**
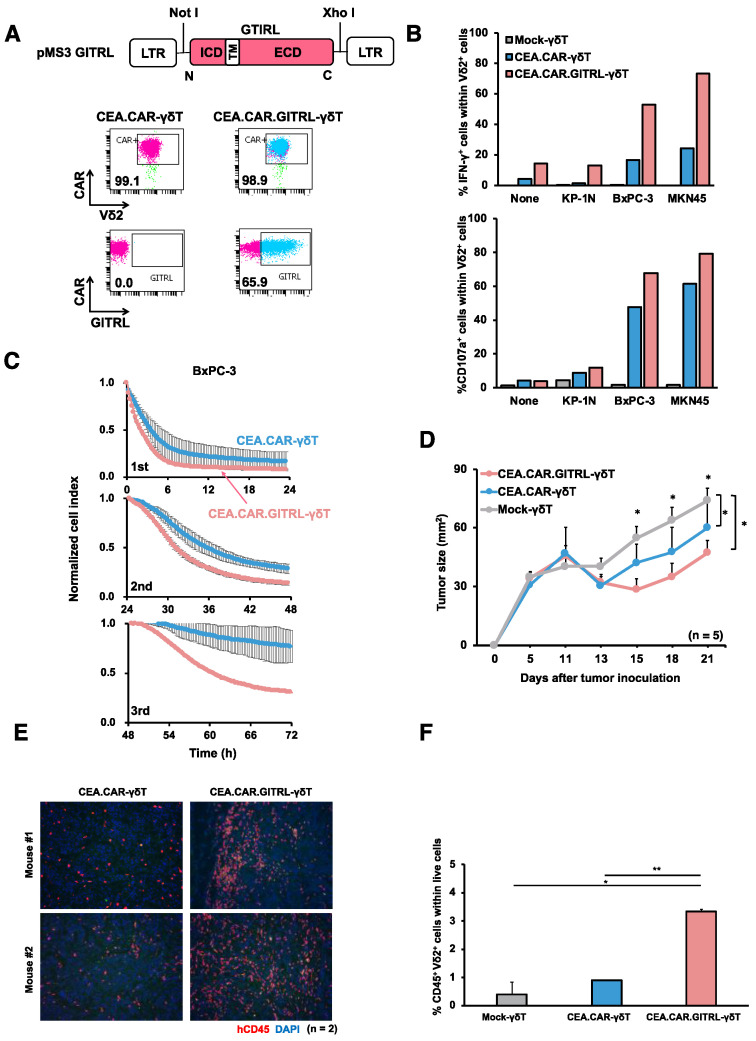
Enhancement of CEA.CAR-γδ T cell function by GITR mediated signaling. (**A**) Schematic representation of a retroviral vector encoding GITR-ligand (GITRL) and FACS profile of GITR ligand expression on CEA.CAR-γδT cells co-transduced with GITRL. Cell surface expression of CAR and GITRL on Vδ2+ cells on day 10. (**B**) CEA.CAR-γδ T cells co-expressing or not-expressing GITRL were stimulated with CEA^+^ (BxPC-3 and MKN45) or CEA^-^ (KP-1N) tumor cell lines and subjected to ICS for IFN-γ and CD107a after gating on Vδ2^+^ cells. (**C**) Serial killing activity of CEA.CAR-γδ T cells co-expressing or not-expressing GITRL was assayed as legend for Figure 2E up to 3 rounds. Each killing was monitored at 24 h of each round. A representative result of 2 independent experiments is shown. Error bars represent SD of the mean. (**D**) Tumor growth curve of NOG mice transferred with CEA.CAR-γδ T cells co-expressing or not-expressing GITRL and Mock-γδ T cells. NOG mice (n = 5) were inoculated s.c. with BxPC-3 (5 × 10^6^ cells) (on day 0) followed by i.v. injection with CEA.CAR-γδ T cells with (CEA.CAR-γδ T) or without GITRL (CEA.CAR.GITRL-γδ T), or Mock-γδ T cells (5 × 10^6^ cells) (on day 7). Tumor areas were measured by a caliper using the formula (length × width) at the indicated time points. Error bars represent SD of the mean. * *p* < 0.05. (**E**) Fluorescence IHC analysis of tumor tissues stained with anti-hCD45 (red) and DAPI (blue) collected from BxPC-3 bearing NOG mice (n = 2) on day 21 after the transfer. A representative result from 3 independent experiments is shown. (**F**) PBMCs from tumor-bearing NOG mice (n = 3) were collected at 21 days after the transfer with (5 × 10^6^) CEA.CAR-γδ T cells, and subjected to flow cytometry analysis. The percentages of Vδ2+ cells obtained using total live cells based on FSC and SSC profiles. Error bars represent SD of the mean. * *p* < 0.05, ** *p* < 0.01.

## Data Availability

There is no electronic datasheet associated with this paper. No data in electronic repository.

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
