# Peer review of "CAR-Modified Vγ9Vδ2 T Cells Propagated Using a Novel Bisphosphonate Prodrug for Allogeneic Adoptive Immunotherapy"

_ijms, 2023, doi:10.3390/ijms241310873_

Round 1
Reviewer 1 Report
- The manuscript's topic is fascinating and with a growing field of application.
- The introduction section is too brief and has fragmented information. The introduction section of the paper must precisely discuss the motivation for the research, well-organized literature review with the latest references, research gap, and contributions of the present work.
- Authors should cite few recent papers in introduction such as:
* Cancer immunotherapy: A comprehensive appraisal of its modes of application (Review)
* Advances in CAR-T Cell Genetic Engineering Strategies to Overcome Hurdles in Solid Tumors Treatment
- The objectives of this paper need to be polished. Contribution list should be polished at the end of the introduction section.
- Please explain the tools and parameters for experiments.
- Include limitations of the study and future work in the to improve the conclusion section.
- Please polish the English writing and also check the typos errors carefully. In addition, please also make sure the paper format is correct.
- Please polish the English writing and also check the typos errors carefully. In addition, please also make sure the paper format is correct.
Author Response
- The manuscript's topic is fascinating and with a growing field of application.- The introduction section is too brief and has fragmented information. The introduction section of the paper must precisely discuss the motivation for the research, well-organized literature review with the latest references, research gap, and contributions of the present work.- Authors should cite few recent papers in introduction such as:
* Cancer immunotherapy: A comprehensive appraisal of its modes of application (Review)
* Advances in CAR-T Cell Genetic Engineering Strategies to Overcome Hurdles in Solid Tumors Treatment
Response: Thank you for the comment. We revised the introduction section according to the comment and cited the suggested papers.
- The objectives of this paper need to be polished. Contribution list should be polished at the end of the introduction section.
Response: Thank you for the comment. We added sentences describing our objective of this study. The contribution list has been modified in the AUTHOR CONTRIBUTION section to clearly indicate who conducted each part of the study.
- Please explain the tools and parameters for experiments.
Response: We apologize for the omission. We added the names of the machines used in data collection and the software used in the data analysis in the Materials and Methods section.
- Include limitations of the study and future work in the to improve the conclusion section.
Response: Thank you for the suggestion. The limitation of the present study that used mainly an ectopic xenograft model in NOG mice and the future work that will use an orthotopic xenograft model in fully humanized mice is discussed in the conclusion of the Discussion section.
- Please polish the English writing and also check the typos errors carefully. In addition, please also make sure the paper format is correct.
Authors’ response: We apologize for the typos. The entire revised manuscript has been checked and corrected by an English speaker. The paper format has been corrected.
Reviewer 2 Report
In this work authors offer insights into a novel approach in cancer immunotherapy using γδT cells as a source for CAR modification. One of the main advantages offered by this approach would be the possibility of allogenic setting which could expediate therapy regimen and possibly achieve prompt, more accessible and efficient therapy option in the future.
The concept of the study is innovative, while experiments conducted in vitro, in vivo and ex vivo produced solid amount of evidence to support it. In addition, up-to-date and comprehensive methodology, with clearly pointed limitations of the study, qualify this manuscript for publication.
Author Response
In this work authors offer insights into a novel approach in cancer immunotherapy using γδT cells as a source for CAR modification. One of the main advantages offered by this approach would be the possibility of allogenic setting which could expediate therapy regimen and possibly achieve prompt, more accessible and efficient therapy option in the future.The concept of the study is innovative, while experiments conducted in vitro, in vivo and ex vivo produced solid amount of evidence to support it. In addition, up-to-date and comprehensive methodology, with clearly pointed limitations of the study, qualify this manuscript for publication.
Authors’ response: Thank you very much for the favorable comments.
Reviewer 3 Report
1) The abstract would benefit from significant revision. I would recommend authors better define the problem they are trying to address and why the issue they are addressing is a problem. After a brief summary of their results, I would recommend discussing why those results are important and how they benefit the field. It would also benefit from review from a native English speaker.
2) Throughout the introduction and manuscript in general, authors use acronyms and abbreviations that have not been defined or are not defined till later. Per standard convention, the first time an acronym/abbreviation is used it should be defined. For example, 'graft versus host disease (GVHD)'. Areas to address in this manuscript include but are not limited to CAR, IFN, MHC, GVHD, etc.
3) In the introduction, readers would benefit from brief descriptions (a phrase is fine) of the topics you are introducing, such as what is butyrophilin, GVHD, why GVHD is a problem/how prevalent it is, why phosphoantigen targeting is a benefit to prevent GVHD, and some details on the benefit of gamma-delta T cells in the clinical trials they cite.
4) Unfortunately, I cannot see any of the Figures for the Results section. Apologies if this is an error on my end, but I have reviewed the manuscript fully as well as the supplementary files and I do not see Figures 1-6. While the results section and figure legends do seem well thought-out, I would need to see the actual figures to give appropriate peer review. For this reason alone I recommend 'Major Revision'. Once this is addressed, I would be happy to review and give additional recommendations.
5) The methods and discussion appear complete, however I would need the Figures to make definitive recommendations.
Thank you for your interesting and exciting paper. I do feel that this manuscript may benefit from review by a native or expert English speaker, particularly in the 'abstract' and 'introduction' sections.
Author Response
1) The abstract would benefit from significant revision. I would recommend authors better define the problem they are trying to address and why the issue they are addressing is a problem. After a brief summary of their results, I would recommend discussing why those results are important and how they benefit the field. It would also benefit from review from a native English speaker.
Response: Thank you for the valuable suggestions. We corrected these issues in the Abstract of the revised manuscript. However, we omitted some of the description due to the word limit (200 words maximum).
2) Throughout the introduction and manuscript in general, authors use acronyms and abbreviations that have not been defined or are not defined till later. Per standard convention, the first time an acronym/abbreviation is used it should be defined. For example, 'graft versus host disease (GVHD)'. Areas to address in this manuscript include but are not limited to CAR, IFN, MHC, GVHD, etc.
Response: Thank you for pointing this out. We defined the acronyms and abbreviations in the revised manuscript.
3) In the introduction, readers would benefit from brief descriptions (a phrase is fine) of the topics you are introducing, such as what is butyrophilin, GVHD, why GVHD is a problem/how prevalent it is, why phosphoantigen targeting is a benefit to prevent GVHD, and some details on the benefit of gamma-delta T cells in the clinical trials they cite.
Response: Thank you for the valuable comment. We described these topics in a concise manner in the Introduction section of the revised manuscript.
4) Unfortunately, I cannot see any of the Figures for the Results section. Apologies if this is an error on my end, but I have reviewed the manuscript fully as well as the supplementary files and I do not see Figures 1-6. While the results section and figure legends do seem well thought-out, I would need to see the actual figures to give appropriate peer review. For this reason alone I recommend 'Major Revision'. Once this is addressed, I would be happy to review and give additional recommendations.
5) The methods and discussion appear complete, however I would need the Figures to make definitive recommendations.
Response: Thank you. We are looking forward to seeing your final recommendation.
Comments on the Quality of English Language
Thank you for your interesting and exciting paper. I do feel that this manuscript may benefit from review by a native or expert English speaker, particularly in the 'abstract' and 'introduction' sections.
Response: We apologize for the language. The entire revised manuscript has been checked and corrected by an English speaker.
Round 2
Reviewer 3 Report
Thank you for your timely and excellent revision. I have no additional concerns.